# The Role and the Effect of Magnesium in Mental Disorders: A Systematic Review

**DOI:** 10.3390/nu12061661

**Published:** 2020-06-03

**Authors:** Andrea Botturi, Valentina Ciappolino, Giuseppe Delvecchio, Andrea Boscutti, Bianca Viscardi, Paolo Brambilla

**Affiliations:** 1Neurologic Clinic, Fondazione IRCCS Istituto neurologico Carlo Besta, 20133 Milan, Italy; 2Department of Neurosciences and Mental Health, Fondazione IRCCS Ca’ Granda-Ospedale Maggiore Policlinico, 20122 Milan, Italy; valentina.ciappolino@gmail.com (V.C.); a.boscutti@gmail.com (A.B.); bianca.viscardi@unimi.it (B.V.); paolo.brambilla1@unimi.it (P.B.); 3Department of Pathophysiology and Transplantation, University of Milan, 20122 Milan, Italy; giuseppe.delvecchio@unimi.it

**Keywords:** magnesium, mental disorders, depression, bipolar disorder, schizophrenia, obsessive-compulsive disorder, autism, anxiety disorder, eating disorder

## Abstract

Introduction: Magnesium is an essential cation involved in many functions within the central nervous system, including transmission and intracellular signal transduction. Several studies have shown its usefulness in neurological and psychiatric diseases. Furthermore, it seems that magnesium levels are lowered in the course of several mental disorders, especially depression. Objectives: In this study, we wish to evaluate the presence of a relationship between the levels of magnesium and the presence of psychiatric pathology as well as the effectiveness of magnesium as a therapeutic supplementation. Methods: A systematic search of scientific records concerning magnesium in psychiatric disorders published from 2010 up to March 2020 was performed. We collected a total of 32 articles: 18 on Depressive Disorders (DD), four on Anxiety Disorders (AD), four on Attention Deficit Hyperactivity Disorder (ADHD), three on Autism Spectrum Disorder (ASD), one on Obsessive–Compulsive Disorder (OCD), one on Schizophrenia (SCZ) and one on Eating Disorders (ED). Results: Twelve studies highlighted mainly positive results in depressive symptoms. Seven showed a significant correlation between reduced plasma magnesium values and depression measured with psychometric scales. Two papers reported improved depressive symptoms after magnesium intake, two in association with antidepressants, compared to controls. No significant association between magnesium serum levels and panic or Generalized Anxiety Disorder (GAD) patients, in two distinct papers, was found. In two other papers, a reduced Hamilton Anxiety Rating Scale (HAM-A) score in depressed patients correlated with higher levels of magnesium and beneficial levels of magnesium in stressed patients was found. Two papers reported low levels of magnesium in association with ADHD. Only one of three papers showed lower levels of magnesium in ASD. ED and SCZ reported a variation in magnesium levels in some aspects of the disease. Conclusion: The results are not univocal, both in terms of the plasma levels and of therapeutic effects. However, from the available evidence, it emerged that supplementation with magnesium could be beneficial. Therefore, it is necessary to design ad hoc clinical trials to evaluate the efficacy of magnesium alone or together with other drugs (antidepressants) in order to establish the correct use of this cation with potential therapeutic effects.

## 1. Introduction

Psychiatric disorders are estimated to affect more than one billion people worldwide [1] and recent reports claimed that the global burden of mental illness accounts for as many as one-third of years lived with disability (YLDs) and 13% of disability-adjusted life years (DALYs) [2]. These figures are associated with a high economic burden since the global direct and indirect economic costs of mental disorders are estimated to be 2.5 trillion USD [3].

The therapeutic options for treating psychiatric disorders are progressively expanding, with both pharmacologic and non-pharmacologic treatments becoming available in recent years. Moreover, although the majority of drugs currently used to treat the more common psychiatric disorders have been proven to be effective by recent robust meta-analyses [4,5], most of the psychiatric pharmacotherapy must be continued for years (or even lifelong), bringing severe adverse side effects [6,7]. Furthermore, the last decade has seen a steep increase in the price of both brand and generic psychiatric prescription drugs [8] and therefore there is a greater need for new therapeutic options that prove to be effective, safe and affordable for the patient and the healthcare system.

Oral nutritional supplements have been shown to improve clinical outcomes of hospitalized patients [9] and to be cost-effective interventions [10]. In recent years, there has been a surge in the number of studies performed in the field of the so-called Nutritional Psychiatry [11], where researchers have been focusing not only on the effects of general dietary modifications on some psychiatric illnesses, in particular mood disorders [12], but also on the potential role of supplementation of single micronutrients in patients with mental disorders [13]. The most substantial pieces of evidence on the efficacy of these interventions are probably those coming from n-3 polyunsaturated fatty acids (n-3PUFAs) supplementation in depression [14,15], which, in a few years, will likely become part of the standard therapy for depression [16].

Among micronutrients, magnesium (Mg^2+^) plays a critical role in brain function and mood since it is essential for optimal nerve transmission and it is involved in the formation of membrane phospholipids. For this reason, it plays a fundamental role in the correct functioning of the central nervous system [17,18].

Specifically, for psychiatric illnesses, multiple studies have been performed in which magnesium serum levels were assessed in cohorts of patients with depression [19], SCZ [20], addiction disorders [21], AD [22] and ADHD [23].

Magnesium has also been studied, both in the form of enriched diet and supplementation at high doses, as an adjunct therapy for psychiatric disorders, in particular in anxiety [24] and mood [25] disorders. However, results from reports on both magnesium levels and supplementation in psychiatric illnesses are often conflicting, potentially due to methodological heterogeneity, which involves, among others, measuring techniques (extracellular vs. ionized magnesium) [26] and supplementation modalities (dose, posology, magnesium form used).

Since magnesium could represent a potentially novel adjunctive therapy in mental disorders, with this review, we aim to give a comprehensive picture of the relationship between magnesium and psychiatric disorders. In particular, we will try to answer the following questions: (a) Are magnesium levels altered in patients with psychiatric disorders? (b) Is magnesium supplementation effective and safe in patients suffering from mental illnesses? If so, is it possible to identify a preferred dose, posology or element form?

To answer these questions, we performed a comprehensive review in which we included all studies that involved patients with a wide variety of psychiatric disorders where magnesium levels were assessed and/or magnesium supplementation was given.

### Biological Plausibility of Magnesium for Brain and Psychiatric Disorders

Magnesium is the fourth most abundant mineral ion, and the intake comes mainly from the ingestion of leafy green vegetables, whole grains, nuts, and fish. Magnesium is absorbed in the gastrointestinal tract and the renal system. This element facilitates calcium (Ca^2+^) absorption and both ions are regulated by the parathyroid hormone; however, free ion concentration does not always correlate with total concentration. Magnesium is primarily found within the cells, while extracellular magnesium accounts for ∼1% of total body magnesium. Furthermore, serum magnesium is present in three forms, with ionized magnesium having the highest biological activity [27].

Magnesium is essential to ensure the correct functioning of all human cells, neurons included; it is involved, among others processes, in hundreds of enzymatic reactions [28], intracellular transmission [29], myelination process [30], synapses formation and maintenance [31] as well as in the regulation of serotoninergic, dopaminergic and cholinergic transmission [32]. Magnesium is, therefore, an element necessary to maintain neurons healthy and viable [33], especially because it has been shown to reduce apoptosis in an animal model of induced hypoxia-ischemia [34] and to prevent synaptic loss in a mouse model of Alzheimer disease [35,36]. Shreds of evidence also suggest that magnesium is involved in neurogenetic processes and the maturation of newly generated neural cells; in fact, magnesium has been proven to efficiently enhance the proliferation of neural stem cells [37] and neurite outgrowth [38]. Through the induction of synaptic plasticity and potentiation of synaptic transmission in the rat hippocampus, magnesium has also been shown to enhance learning abilities, working memory and short- and long-term memory [39].

The antidepressant action of magnesium is likely to be mediated by several mechanisms. The most important one seems to be the one involving the blockade of the glutamatergic N-methyl-D-aspartate receptor (NMDAR); interestingly, this is the same target of the fast-acting antidepressant ketamine, an NMDAR antagonist as well. However, other components of glutamatergic transmission, such as the α-amino-3-hydroxy-5-methyl-4-isoxazolepropionic acid AMPA receptor, appear to be modulated by magnesium as well [40]. Another relevant finding that supports the notion of magnesium antidepressant activity is the one linking magnesium deficiency to dysregulation in the hypothalamic–pituitary–adrenal (HPA) axis, which is well known to be involved in the pathogenesis of anxiety disorders and depression [41]. Conversely, increased levels of brain magnesium have been shown to enhance (a) the retention of the extinction of fear memory, through increased NMDA signaling, (b) the brain-derived neurotrophic factor (BDNF) expression and (c) synaptic plasticity in the prefrontal cortex (PFC); notably, these effects were absent in another region closely linked to depression pathogenesis, such as the basolateral amygdala [42].

The antidepressant action of magnesium appears to be, at least partially, mediated by a modulation of the serotoninergic system; in fact, it seems that magnesium has a synergistic effect when administered with molecules of the selective serotonin reuptake inhibitor (SSRI) class and that the antidepressant action of magnesium is impaired when animals are pre-treated with a compound that inhibits the serotonin synthesis [43]. Interestingly, in rats, a diet deficient in magnesium was also found to be associated with alterations in the gut microbiota, ultimately leading to alterations in the gut-brain axis and the development of depressive-like behaviors [44]. Finally, Sowa-Kuæma et al. [45] found that concentrations of both magnesium and zinc (Zn) were reduced in the hippocampal tissue of suicide victims, along with an altered glutamatergic NMDA activity in the hippocampus. Moreover, several reports suggest that magnesium is a key mediator of the efficacy of antipsychotic medications. Both haloperidol and risperidone were found to increase intra-erythrocytic magnesium levels [46] and treatment of hypoparathyroidism-induced psychosis seems to be dependent on magnesium level, with hypomagnesemia causing treatment resistance to antipsychotics [47]. Magnesium also appears to be involved in both prevention and reversal of movement disorders induced by the chronic use of typical antipsychotics. Moreover, in an animal model, magnesium was found to reduce the severity of movement disorders via the prevention of the formation of reactive oxygen species in cortical areas, striatum and substantia nigra [48].

## 2. Materials and Methods

A comprehensive search of all studies using or analyzing the effect of magnesium in psychiatric disorders were conducted on the PubMed database from 1 January 2019 to 30 March 2020. The search was re-run on a weekly basis, with the last search performed on 10 April 2020. Finally, we searched for potentially valuable records by scanning reference lists of articles relevant to the topic. The final search strategy for PubMed was designed by AB and the search syntax is reported in the Appendix A. 

Articles of potential interest were identified by using the following search strategy: “(magnesium OR Mg^2+^) AND (psychiatric disorders OR mental diseases OR psychotic disorders OR psychosis OR ultra-high risk psychosis OR schizophrenia OR bipolar disorder OR affective disorder OR major depressive disorder OR depression OR personality disorder OR anxiety disorders OR obsessive compulsive disorders OR eating disorders OR ADHD OR autism)”. Only studies in English (or with an English translation available) were taken into consideration. Relevant articles were obtained and included in the review if (a) they reported an exposure to magnesium, (b) included psychiatric symptoms as an outcome measure and (c) enrolled human participants and reported a trial or an observational study. We considered cohort studies by exploring serum levels of magnesium as primary outcome and trials in which the authors used an exposure of magnesium alone or as an adjunctive therapy to other drugs (e.g., antipsychotic, antidepressants, mood stabilizers and benzodiazepines), or other non-pharmacological strategies, such as psychotherapy and physical exercise compared to placebo or pharmacotherapy alone. In addition, we included trials that employed a diet enriched in magnesium as a supplementation. All the studies, both clinical trials and cohort studies (prospective or retrospective), reporting the effects of magnesium in patients were included.

To limit the heterogeneity of this review and to reduce selection bias, we decided to exclude: (1) studies that did not explore the effects of magnesium on psychiatric symptoms as primary outcome; (2) pre-clinical studies, both in vitro and in vivo (animal); (3) case-control, case series or case reports; (4) reviews and/or metanalyses.

Search results were exported into the reference manager software “Rayyan QCRI”; duplicates detected by the software were resolved manually by (AB). After duplication removal, all the resulting records were screened by title and abstract by AB and BV and initially labeled for inclusion with “included”, “excluded” or “maybe”. Inclusion of records labeled with “maybe” was discussed between the two reviewers, with other members of the team involved if needed. In the second level of screening, full text of publications was evaluated by AB and BV; disagreements on study selection were resolved by consensus with the involvement of a third author (VC). Two reviewers (AB and BV) independently charted the data and discussed the results; any disagreement was resolved by consensus with the involvement of a third team member (VC). Since the search was re-run on a weekly basis, data from newly included study were updated accordingly. Then data items collected were analyzed according to the following variables that were abstracted for each article (see Appendix A). 

The results were grouped based on whether they were observational or clinical trials investigating the effect of magnesium on patients’ mental health. Findings were discussed on the basis of different diagnoses made (e.g., depression, schizophrenia) and are summarized in table form. The results for each clinical topic are presented in a narrative form in the main text. 

We identified 1104 citations by searching the PubMed database. After duplicates were removed, a total of 698 records were considered for inclusion. After the first screening, based on the title and abstract, 621 records were excluded, with 77 full text articles to be retrieved and assessed for eligibility. After this second screening process, 45 studies were excluded for the following reasons: 11 were animal studies; seven did not report the effect of magnesium as primary outcomes; 18 did not report as results the serum levels of magnesium; three were reviews or metanalyses; four were preclinical studies and two did not investigate an adult sample. Finally, a total of 32 articles were considered eligible for this review. The process of the identification and inclusion of trials is summarized in Figure 1 (PRISMA diagram). 

Relevant articles were obtained and included in the review if (a) they reported an exposure to magnesium, (b) included psychiatric symptoms as an outcome measure, and (c) enrolled human participants and reported a trial.

## 3. Results

Most of the results of the reviewed studies focused on depression and depressive symptoms and only a small number of studies concerning other psychiatric disorders were found.

### 3.1. Depression

Risk factors for depression include dietary patterns. Some epidemiological or observational studies reported that greater dietary intake of magnesium is linked to a general reduced risk of depressive disorders or fewer depressive symptoms [50,51]. However, some studies also suggested that, in depressive disorders, magnesium plasma levels can show different or synergistic effects [52]. To clarify this issue, in this paragraph we reviewed all randomized clinical trial (RCT) studies, exploring the impact of magnesium on depressive disorders. Firstly, we explored the RCT studies (Table 1) that explored magnesium levels in depressed patients. We identified twelve studies showing mainly positive results, even though they were conducted in different populations and measured symptoms with various tools.

A group of studies investigated several microelements, including serum magnesium, in depressed patients vs. healthy controls. Most studies reported a significant decrease in concentrations of magnesium in depressed patients [53,54,56,57,58] whereas two studies [59,60], conducted in a female population, reported no significant decrease in magnesium levels.

We identified three studies [61,62,63] that correlated magnesium levels with the severity of symptoms, measured with a variety of tools, but reporting mixed results.

One study [61] showed only a small correlation between psychomotor retardation and magnesium plasma levels. Interestingly, when patients were divided between responder and non-responder to treatment (SSRI or SNRI), they demonstrated that patients with higher plasma magnesium levels at baseline improved more compared to those with lower magnesium levels at baseline.

Finally, we identified two studies conducted on healthy populations, evaluating the correlation between magnesium levels and depressive symptoms. Specifically, Tarleton et al. (2019) [64] considered Patient Health Questionnaire (PHQ) scores in a large cohort of 3604 healthy adults and showed a significant relationship between serum magnesium and symptoms. Similarly, Salehi-Pourmehr et al. (2019) [65], who conducted a study on overweight pregnant women, also reported a positive significant correlation.

Furthermore, we identified five RCT studies (Table 2) in which magnesium was administered alone or as an add-on treatment to depressed patients, also reporting contrasting results. Specifically, we found three positive studies showing the efficacy of magnesium supplementation in the treatment of depression [77,78,82]. In contrast, Fard et al. (2017) [80] showed that magnesium did not reduce anxiety and depressive symptoms in postpartum women. In the same direction, Mehdi et al. (2017) [83] did not find a significant effect of magnesium sulfate in affecting depression.

Finally, a single, smaller study [79] addressing the efficacy and safety of magnesium as an augmentation to antidepressant treatment, found no significant differences.

In conclusion, current evidence on the impact of magnesium on depression should be supported using longitudinal studies with extended follow up, larger sample sizes and repeated psychopathological evaluations at different times.

### 3.2. Other Psychiatric Disorders

#### 3.2.1. Anxiety Disorders

In our record screening process, we identified four studies that analyzed the concentration of magnesium in patients suffering from anxiety disorders. Three of them reported no significant differences in magnesium serum levels in Generalized Anxiety Disorder (GAD), Panic Disorder and anxiety symptoms during a major depressive episode [55,61,66]. In contrast, in Camardese et al. (2012) [61], the authors hypothesized that hypomagnesaemia could play a role in drug responsiveness among depressed patients, as they found a correlation between lower magnesium levels and poor outcomes in treated patients. 

Garalejić et al. (2010) [67] also investigated the relationship between Hamilton Anxiety Scale (HAMA) scores and magnesium levels, although, unlike other studies identified, this study considered magnesium levels in peritoneal fluid among 87 infertile women undergoing laparoscopy, finding a strong negative correlation between the severity of anxiety symptoms and magnesiumg concentration in peritoneal fluid. Therefore, the authors hypothesized that the decrease in magnesium peritoneal concentration may be caused by the higher production of endogenous catecholamines (predominantly adrenaline) in patients showing higher scores at the anxiety symptoms assessment.

Finally, as for the use of magnesium supplementation in psychopharmacological treatment, we found one study that considered the effects of magnesium supplementation on anxiety symptoms [80]. As depicted above, Fard et al. (2017) [80] explored both the baseline trace element level and possible benefits of Zn and Mg^2+^ supplements on depression and anxiety symptoms among postpartum women. Anxiety symptoms were evaluated using the Spielberger State–Trait Anxiety Inventory and no statistically significant difference was observed in mean scores of state anxiety and trait anxiety.

#### 3.2.2. Obsessive–Compulsive Disorder (OCD)

There are very few scientific data about serum levels of microelements in Obsessive–Compulsive Disorder (OCD) patients. Indeed, we found only one study carried out by Shohag et al. (2012) [68] that described a decrease in magnesium levels, together with Zn and Fe levels, in OCD patients when compared to healthy controls.

#### 3.2.3. Schizophrenia

We found a total of one study concerning Mg and SCZ (Table 1). Specifically, Jabotinsky-Rubin et al. (1993) [84] reported that patients with SCZ had increased magnesium plasma concentrations and also that magnesium levels were reduced after the administration of haloperidol. In contrast, Athanassesnas et al. (1983) [85] reported no differences in plasma magnesium concentrations in drug-free patients with SCZ. Moreover, in our record screening process, we found one study, carried out by Ruljancic et al. (2013) [69], that studied magnesium concentration in suicidal and non-suicidal patients with SCZ, which reported a higher magnesium concentration in the platelets of suicidal patients and a higher Ca^2+^/Mg^2+^ ratio in the platelets of non-suicidal patients, an indirect index of higher Ca^2+^ concentration. However, the imbalance found in the two electrolytes in the platelets of suicidal and non-suicidal patients with SCZ still requires further research in order to clarify the involvement of Mg and Ca^2+^ in SCZ, as well as a possible relation to antipsychotic action.

#### 3.2.4. Eating Disorders

A large retrospective study by Raj et al. (2012) [70] aimed to determine the prevalence of hypomagnesemia (Mg^2+^ ≤ 1.7 mg/dL) and clinical characteristics of adolescents hospitalized with a DSM-IV [86] diagnosis of eating disorder. As expected, they found hypomagnesemia in approximately 16% of eligible participants. Furthermore, the authors found that, compared to those with normal serum magnesium levels, patients with hypomagnesemia were older, with longer illness durations, more likely to be purging, and more likely to have an alkaline urine. However, they did not differ in eating disorder diagnosis, Body Mass Index, or other electrolyte disturbances.

#### 3.2.5. Attention Deficit Hyperactivity Disorder (ADHD)

In recent years, some studies [87] have explored the association between micronutrient dietary intake and ADHD development. Specifically, Rucklidge et al. (2019) [81] (Table 1) conducted a parallel group RCT to assess the efficacy and safety of a broad-spectrum micronutrient formula compared with a placebo in 93 medication-free children diagnosed with ADHD. With regard to micronutrient levels among the subjects, most children entered the trial with nutrient blood levels falling within expected ranges. Data analysis only resulted in statistically insignificant associations between micronutrient treatment response and pre-treatment serum nutrient levels, showing the limited value of using serum nutrient levels to predict treatment response. 

Moreover, Yang et al. (2019) [71] (Table 1) conducted a study with the aim of investigating the trace element status of Zn, Cu, Fe, Mg^2+^ and lead in children with ADHD and healthy controls. They enrolled Chinese children diagnosed with ADHD, according to DSM-5 criteria [88], under the following three presentations: predominantly inattentive presentation, predominantly hyperactive/impulsive presentation and combined presentation; they were between the ages of 6 and 16 years and had no history of psychopharmacological treatment for their condition. The authors found that there were alterations in the blood levels of Zn in ADHD patients, which were associated with their symptom scores. However, unlike Zn levels, the study failed to find a correlation between magnesium and ADHD diagnosis or symptom scores.

In contrast, in a case-control study, Mahmoud et al. (2011) [72] found that magnesium levels were significantly lower in children with ADHD compared to controls. Finally, in a very recent publication, Skalny et al. (2020) [73] also observed a significant difference in magnesium between ADHD patients and gender- and age-matched neurotypical controls. However, the patterns of trace element and mineral levels in ADHD were significantly affected by gender and age.

#### 3.2.6. Autism Spectrum Disorder (ASD)

Regarding the role of trace elements in this group of disorders, two studies did not find a statistically significant difference in levels of magnesium in children diagnosed with Autism Spectrum Disorder (ASD) compared to age-matched and gender matched neurotypical children [74,75] (Table 1), while one study [76], demonstrated lower levels of Mg in a large cohort of Chinese children diagnosed with ASD.

## 4. Discussion

The aim of this review was to provide a comprehensive overview of the effects of magnesium in different psychiatric disorders. Interestingly, from the reviewed studies, it emerged that the results showing an association between mental disorders and magnesium are still largely inconclusive. Specifically, we found a great number of studies evaluating serum levels of magnesium in different mental disorders, especially depression. However, only few RCTs tested the efficacy of magnesium alone or as an added therapy in the treatment of different psychiatric disorders and only two studies explored the presence of magnesium in the dietary habits of a schizophrenic and depressed population, respectively. Notably, the presence of many studies on depression is not surprising since the use of magnesium is mostly reserved to depressive disorders, because of its involvement in several core mechanisms of depressive physiopathology, including glutamatergic transmission in the limbic system and cerebral cortex [89], regulation of the HPA axis [41,90], inflammation and oxidative stress [91], response to NMDA receptor antagonist [92], serotonin, dopamine, and noradrenaline modulation [43], BDNF expression [40], as well as the modulation of the sleep–wake cycle [93]. Moreover, previous evidence reported that the potential efficacy of magnesium in depression may be linked to the modulation of glutamatergic signals, which play a key role on neuroprotection, and to the fact that magnesium acts as antagonist to NMDA receptors [42]. Moreover, evidence also showed that magnesium may have a synergistic effect when combined with antidepressants. Indeed, the review carried out by Serefko et al. (2016) suggested that magnesium could improve the efficacy of standard antidepressant treatments, and as such, could be an add-on treatment to the standard antidepressant [94]. The role of magnesium in depression has been also demonstrated in several preclinical studies. Interestingly, Poleszak et al. (2005) found that magnesium enhanced the antidepressant effect of imipramine in mice using a forced swim test (FST) [95]. Additionally, the same research group, in 2006, showed that combining sub-therapeutic doses of Mg^2+^ in combination with sub-therapeutic doses of imipramine leads to a significant antidepressant-like effect in animal models [96]. Furthermore, Singewald et al. (2004) demonstrated that imipramine could reverse depression-like behavior in rats with low levels of magnesium [97]. In addition, Poleszak et al. (2007), demonstrated that magnesium administered in combination with an NMDA antagonist called MK-801, which is similar to ketamine, amplified its antidepressant effect [98]. More recently, Murck et al. (2013) reported that magnesium and ketamine showed an overlap of action in animal models because of both of them could lead to synaptic sprouting [99], ultimately suggesting that they both have a similar action in SNC. Thus, the authors suggested that, in depressed patients, magnesium levels could be used to predict the effect of ketamine [99].

Therefore, based on this evidence, showing the key role of magnesium in influencing mechanisms that may lead to depression, further studies investigating the impact of antidepressant drugs on intracellular magnesium concentration in neurons are required.

Interestingly, the research carried out on depression highlighted the linked between the development of this disabling illness and reduced plasmatic levels of magnesium, evidence that is line with previous reviews and meta-analyses [25]. In the light of these results, several studies [54,56,64] suggested that, for adults seen in primary care, lower serum magnesium levels were associated with depressive symptoms, ultimately supporting the use of supplemental magnesium as a therapy. For this reason, magnesium could be considered a hallmark of pathology or could represent a biomarker of response to drug treatment in patients with mood disorders, as also reported by a previous review [79]. Indeed, patients with therapy refractory depression appear to have lower central nervous system magnesium levels in comparison to health controls [61]. In an attempt to develop nutrition therapies for depressed patients with lower levels of magnesium, it could be useful to use a dose of this mineral combined with the standard antidepressant treatment to ameliorate the outcomes of the disease, providing a personalized approach to depression.

However, only a handful of studies investigated the efficacy of magnesium supplementation alone or as an add-on therapy to other drugs In particular, it emerged from these studies that magnesium alone [77,78,83], magnesium and other micronutrients [80], magnesium with vitamin B6 [82] or magnesium in combination with antidepressants [79] could not be considered significantly effective for treating depression, since the results are conflicting. Interestingly, Medhi et al. (2017) [83], was the only study that administered an intravenous infusion of magnesium and reported that this formulation had only a partial, but insignificant, anti-depressive effect in depressed patients. A possible explanation could be due to the particular pharmacokinetics of magnesium. Indeed, the total concentration of magnesium is mainly intracellular and free ion concentration does not always correlate with whole concentration.

Importantly, comorbidities and other confounders, such as age and geographic location, may contribute to the discrepant findings. In addition, other factors linked to psychiatric patients could impact on these findings, such as sedentary lifestyle, unhealthy dietary patterns, smoking, alcohol or substance abuse and lower compliance with treatments.

Similarly, the evidence reporting the relationship between anxiety disorders and magnesium are still conflicting, although this association is well established in the scientific literature [100]. However, these negative results might be due to the small number of studies investigating magnesium values in anxiety disorders in the past 10 years, ultimately suggesting the need for future research focusing on elucidating magnesium’s mechanism of action in order to determine if it has anxiolytic properties.

Furthermore, mixed results have also been reported by studies investigating the link between dietary pattern and the deficiency of magnesium or other minerals (e.g., Zn, Fe) in ADHD patients, with some studies showing an efficacy and other not. However, it is possible that these heterogenous results could be linked to the different minerals supplemented, meaning that we were unable to examine the real effect of each mineral, and also to different characteristics of patients enrolled in the original studies, which were not homogeneous in terms of age, severity and subtype of ADHD.

Finally, due to the paucity of studies investigating ASD, SCZ and OCD, at present is not possible to determine the role of magnesium in either the physiopathology or in the treatment of these disorders. Similarly, regarding eating disorders, we found only one study where the presence of various internal conditions seems to mask the pathophysiological role of magnesium. This represents an unexplored field and could be interesting in elucidating the role and effect of magnesium in these disorders.

Importantly, this review might be considered in the light of some limitations derived from the heterogeneity of the included studies in terms of (1) types of magnesium supplementation, (2) target population (3), follow-up period, (4) outcome measures, (4) the severity of the illness (5) sample sizes, (6) comorbidities and (7) lifestyle. All these factors may have limited the generalizability of the results and made it difficult to compare the results emerged from the available studies.

In conclusion, due to the lack of consistency between the available studies, there is limited evidence that magnesium, alone or as an add-on therapy, is useful in treating different psychiatric disorders, even if a large amount of data showed reduced plasma levels, especially in depressive patients. Therefore, larger and more homogenous studies are required for showing the role and the effects of magnesium in psychiatric illnesses.

## Figures and Tables

**Figure 1 nutrients-12-01661-f001:**
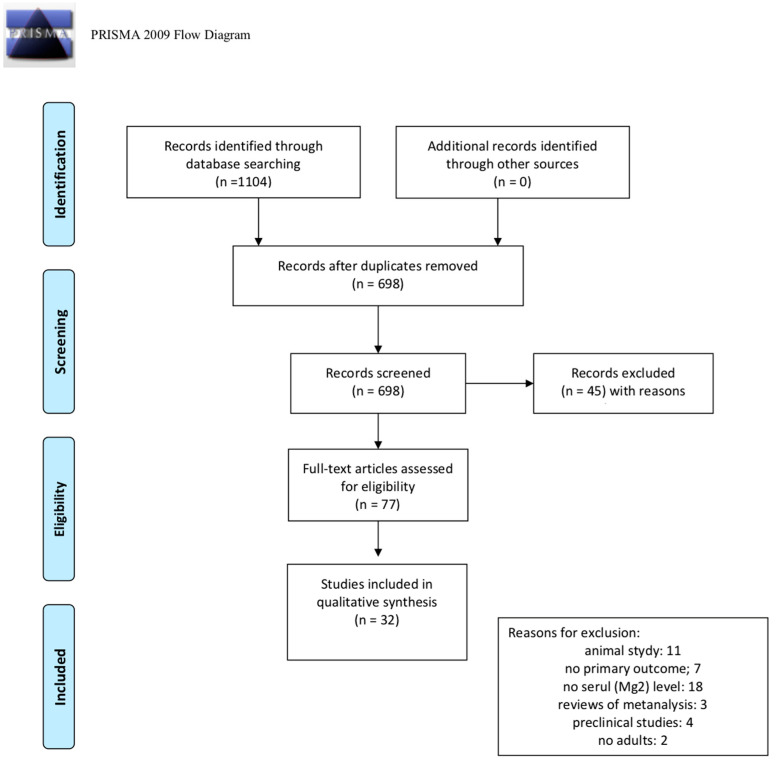
PRISMA (2009) flow diagram [49].

**Table 1 nutrients-12-01661-t001:** Studies of magnesium levels.

Author, Year	Title	Sample	Study Type	Psychiatric Disorder	Psychopathological Scale	Results	Direction of Evidence
[53]	Total and ionized calcium and magnesium are significantly lowered in drug-naïve depressed patients: effects of antidepressants and associations with immune activation	(a) 140 MDD + 40 hc; treatment in 48 MDD; (b) 44 patients, 2 months with blood samplings (baseline and during treatment)	(a) case-control study; (b) prospective study	Depression	BDI-II	In MDD patients Serum Ca and Mg (total and ionized) were significantly lower compared with controls. Antidepressants increased Ca and lowered Mg levels. Significant and inverse correlations between the BDI-II scores from baseline to endpoint and Ca (both total and ionized), but not Mg levels. Antidepressants probably reduced Mg levels as a side effect	(+)
[54]	Biomarkers in Drug Free Subjects with Depression: Correlation with Tryptophan	96 (48 depression, 48 controls)	cross-sectional study	Depression	HAMD	Depression is associated with deficiency of TRP, Se, Vit D, Mg. (The association among TRP and other biomarkers is non-significant)	(+)
[55]	Comparative analysis of serum zinc, copper, manganese, iron, calcium, and magnesium level and complexity of interelement relations in generalized anxiety disorder patients	50 GAD, 51 hc	comparative study	GAD	DSM-IV criteria	Ca and Mg concentration between patient and control groups were not significant (*p* > 0.05)	(−)
[56]	Alterations of serum macro-minerals and trace elements are associated with major depressive disorder: a case-control study	247 patients and 248	prospective case-control study	Depression	DSM-5 criteria	Decreased concentrations of Ca and Mg, Fe, manganese, selenium, and zinc in MDD patients compared with control subjects. Data obtained from different inter-element relations in MDD patients and control subjects strongly suggest that there is a disturbance in the element homeostasis	(+)
[57]	Serum Vitamin D and Magnesium levels in a psychiatric cohort	73 psychiatric day treatment unit cohort	cross-sectional analysis	Miscellaneous	ICD-10 criteria	The percentage of patients who were magnesium deficient was 78.6% (*n* = 22/28)	(+)
[58]	Depression in elderly type 2 diabetes	210 type 2 diabetes patients aged 65 years and above	cross-sectional study	Depression in elderly type 2 diabetes	DSM-IV criteria	Among all patients, 88.6% had magnesium intake which was less than the dietary reference intake, and 37.1% had hypomagnesaemia. The odds of depression, central obesity, high body fat percentage, and high body mass index were significantly lower with increasing quartile of magnesium intake (*p* for trend < 0.05). The majority of elderly type 2 diabetes who have low magnesium intake may compound this deficiency with metabolic abnormalities and depression	(+)
[59]	Low dietary calcium is associated with self-rated depression in middle-aged Korean women	105 women age 41–57; 51 pre-meno-pausal, 54 post-meno-pausal	case-control study	Depression in middle-aged women	SDS	No significant differences in serum levels of Ca and Mg among the three groups with different severity of symptoms. Negative correlations between SDS and Ca intake and animal Ca after adjusting for age, menopause and energy intake	(−)
[60]	Correlation between Depression with Serum Levels of Vitamin D, Calcium and Magnesium in Women of Reproductive Age	100 women 15–44 years old	cross-sectional study	Depression	BDI-II	Women’s depression scores showed a significant inverse correlation with the serum level of vitamin D (*r* = -0.21, *p* = 0.03). No significant correlations with serum levels of calcium and magnesium	(−)
[61]	Plasma magnesium levels and treatment outcome in depressed patients	123 outpatients	observational study	Depression	HAMD; HAMA; (DRRS); (SHAPS)	No association between Mg levels and psychopathological severity Patients who responded to antidepressant treatment showed higher Mg levels and higher retardation scores at basal evaluation in comparison with non-responders	(−)
[62]	Variability in serum electrolytes in different grades of depression	100 MDD (age 35–45), 100 hc matched)	cross-sectional study	Depression	DSM-IV, ICD-10 criteria; HAMD	All the depression patients were having higher level of Na, K, and Ca and lower level of Mg. Multiple comparison revealed highly statistically significant difference between the levels of serum Ca and Mg in three levels of severity (mild, moderate and severe depression)	(+)
[63]	Analysis of Relations Between the Level of Mg, Zn, Ca, Cu, and Fe and Depressiveness in Postmenopausal Women	198 healthy post-meno-pausal women (age 56.26 ± 5.55)	cross-sectional study	Depression	Depressive symptoms in postmenopausal women	Women with depressive symptoms had the lowest Mg levels (14.28 ± 2.13 mg/l), the highest in women without depressive symptoms (16.30 ± 3.51 mg/L), (*p* ≤ 0.05). Authors indicate a higher vulnerability to depression in a group of women with lower levels of Mg and higher levels of Cu	(+)
[64]	The Association between Serum Magnesium Levels and Depression in an Adult Primary Care Population	3604 adults	cross-sectional analysis	Depression	PHQ	The relationship between serum magnesium and depression using univariate analyses showed a significant effect when measured by the PHQ-2 (−0.19 points/mg/dL; 95% CI −0.31, −0.07; *p* = 0.001) and the PHQ-9 (−0.93 points/mg/dL; 95% CI −1.81, −0.06; *p* = 0.037). This relationship was strengthened after adjusting for covariates	(+)
[65]	Screening depression in overweight and obese pregnant women and its predicts	232 overweight or obese pregnant women	cross-sectional study	Depression in pregnancy	Edinburgh Postnatal Depression Scale	Protein, fat, magnesium had positive significant correlation with depression	(+)
[66]	Comparative analysis of serum manganese, zinc, calcium, copper and magnesium level in panic disorder patients	54 panic disorder. + 52 hc	comparative analysis	Panic disorder	DSM-IV criteria	Serum concentration of Zn decreased significantly (*p* = 0.001) in patient group. Otherwise concentration of Mn, Ca, Cu, and Mg were not significant.	(−)
[67]	Hamilton anxiety scale (HAMA) in infertile women with endometriosis and its correlation with magnesium levels in peritoneal fluid	40 endo-metriosis, 47 hc undergoing laparo-scopy (other causes of infertility)	prospective study	Anxiety disorders	HAMA	In infertile women without endometriosis there was a correlation between Mg concentration in peritoneal fluid and HAMA score. No such correlation was found in the women with endometriosis	(+)
[68]	Alterations of serum zinc, copper, manganese, iron, calcium, and magnesium concentrations and the complexity of interelement relations in patients with obsessive-compulsive disorder	48 OCD + 48 hc	cross-sectional study	OCD	Yale-Brown Obsessive Compulsive Scale (YBOCS)	In patients’ serum, zinc, iron, and magnesium concentrations decreased significantly (*p* < 0.05) compared to the controls	(+)
[69]	Platelet and serum calcium and magnesium concentration in suicidal and non-suicidal schizophrenic patients	23 schizophrenics (ICD-10) with attempted suicide + 48 without suicidal behavior + 99 hc	cross sectional study, with 3 groups used for comparison	Suicidal and non-suicidal schizophrenic patients	semi-structured interview (ICD-10 criteria);	A higher Ca/Mg ratio in the platelets of non-suicidal patients confirms indirect higher Ca concentration. Higher Mg concentration in the platelets of suicidal patients, considered a Ca antagonist, may represent a compensatory attempt to restrain Ca activity	(±)
[70]	Hypomagnesemia in adolescents with eating disorders hospitalized for medical instability	541 hospitalized adolescents aged 10–21 years with an eating disorder from 2007 to 2010	retrospective study	Eating disorders	DSM-IV criteria; clinical characteristics	15.9% developed hypomagnesemia. Compared with those with normal serum magnesium levels, patients with hypomagnesemia were older (*p* = 0.0001), ill longer (*p* = 0.001), more likely to be purging (*p* = 0.04), and more likely to have an alkaline urine (*p* = 0.01). They did not differ in eating disorder diagnosis, BMI, or other electrolyte disturbances. Hypomagnesemia is prevalent in adolescents hospitalized for an eating disorder and is associated with purging and alkaline urine	(±)
[71]	Blood Levels of Trace Elements in Children with Attention-Deficit Hyperactivity Disorder: Results from a Case-Control Study	419 ADHD, 395 hc	case-control study	ADHD	Vanderbilt ADHD Diagnostic Parent and Teacher Rating Scales, Conners’ Parent and Teacher Rating Scales (Chinese version), SNAP-IV, Raven’s Progressive Matrices.	Lower zinc levels (*p* < 0.001) and the number out of normal ranges (*p* = 0.015) were found in children with ADHD when compared with the normal control group. The difference remained when adjusting the factor of BMI z-score. No significant between-group differences were found in levels of other elements	(+)
[72]	Zinc, ferritin, magnesium and copper in a group of Egyptian children with attention deficit hyperactivity disorder	58 ADHD, age 5–15 + 25 hc, matched	case-control study	ADHD	Conner’s Rating Scales, discriminating between children with ADHD and h.c., as well as severity of ADHD.	Zinc, ferritin and Mg levels were significantly lower than controls (*p* value 0.04, 0.03 and 0.02 respectively). No significant differences between sub-groups of ADHD	(+)
[73]	Serum zinc, copper, zinc-to-copper ratio, and other essential elements and minerals in children with attention deficit/hyperactivity disorder (ADHD)	136 (children) (68 ADHD, 68 hc, matched)	cross-sectional study	ADHD	CD-10 criteria (F90.0)	Cr, Mg, and Zn levels in children with ADHD were 21% (*p* = 0.010), 4% (*p* = 0.005), and 7% (*p* = 0. 001) lower as compared to the healthy controls, respectively. However, the patterns of trace element and mineral levels in ADHD were significantly affected by gender and age. Hypothetically, the observed decrease in essential trace elements, namely Mg and Zn, may significantly contribute to the risk of ADHD or its severity and/or comorbidity	(±)
[74]	Evaluation of Whole Blood Trace Element Levels in Chinese Children with Autism Spectrum Disorder	113 ASD children, 141 age- and gender-matched neurotypical children		ASD		No significant differences in the whole blood Cu, Zn/Cu ratio, Fe, or Mg was detected between the ASD group and the control group	(−)
[75]	Plasma concentrations of the trace elements copper, zinc and selenium in Brazilian children with autism spectrum disorder.	23 ASD	cross-sectional study	ASD	DSM-5 criteria	The cohort did not show a marked difference in micro-nutrient intake in relation with their resident geographical area and their dietary habit or metabolic state; a slight difference in the levels of magnesium and phosphorus was retrieved due to sex difference	(−)
[76]	Vitamin and mineral status of children with autism spectrum disorder in Hainan Province of China: associations with symptoms.	274 ASD, 97 age-matched hc (typically developed)	Interventional study	ASD	DSM-5 criteria; (ABC), (SRS), (GDS)	The levels of Ca, Mg, Fe, and zinc in children with ASD were significantly lower than those in TD children	(+)

Captions: (MDD) Major depressive disorder; (hc) Healthy controls; (BDI-II) Beck Depression Inventory; (HAMD) Hamilton Depression Rating Scale; (TRP) tryptophan; (GAD) Generalized Anxiety Disorders; (ICD-10) International Classification of Diseases, Tenth Revision; (SDS) Self-rating Depression Scale; (HAMA) Hamilton Anxiety Rating Scale; (DRRS) Depression Retardation Rating Scale; (SHAPS) Snaith - Hamilton Pleasure Scale; (PHQ) Patient Health Questionnaire; (EPDS) Edinburgh Postnatal Depression Scale; (ABC) Autism Behavior Checklist; (SRS) Social Responsiveness Scale; (GDS) Gesell Developmental Scale positive outcome (+); negative outcome (−); unclear result (±).

**Table 2 nutrients-12-01661-t002:** Socio-demographic and clinical details of all included studies investigating the effect of Mg as treatment.

Author, Year	Sample	Study Type	Psychiatric Disorder	Psycho-Pathological Scale	Treatment	Treatment Duration	Results	Outcome	Direction of Evidences
[77]	30 patients (7 M, 19 F; mean age 32.20 ± 9.54) + 30 hc (7 M, 20 F; mean age 32.07 ± 7.69)	Double-blind, placebo-controlled trial	Depression	BDI-II	Mg oxide 250 mg/die	8 weeks	BDI score significantly declined in patients treated with Mg compared to placebo.	Mg supplementation was effective on depression status in depressed patients with Mg deficiency	(+)
[78]	55 randomized to Immediate treatment (22 M, 33 F; mean age 55.2 ± 12.3) + 57 randomized to Delayed treatment (22 M, 35 F; mean age 50.1 ± 13.0)	Randomized case-control clinical trial	Depression	PHQ-9, GAD-7	248 mg of elemental Mg/die	6 weeks	Clinically significant net improvement in PHQ-9 scores of -6.0 points and (GAD-7 in scores of -4.5 points. Similar effects were observed regardless of age, gender, baseline severity of depression, baseline Mg level, or use of antidepressants. Effects were observed within two weeks.	Mg is effective for mild-to-moderate depression in adults	(+)
[79]	17 patients (6 M, 11 F; mean age 48.1±15.5) + 20 hc (10 M, 10 F; mean age 49.7±12.3)	Placebo-controlled study and review	Depression	HAMD, HAMA, CGI	Fluoxetine (20 to 40 mg/die) treatment was augmented with either placebo or Mg 40 mg ×3/die (equivalent to 3.30 mEq of Mg-aspartate)	8 weeks	Fluoxetine + Mg group showed improvement in HDRS scores at week 8 than the fluoxetine+placebo group, but the difference was not statistically significant.	No significant superiority of Mg augmentation therapy on depressive and anxiety symptoms	(±)
[80]	99 women (3 groups, mean age 29.4 ± 5.4, 26.4 ± 4.8, and 27.6 ± 5.1 respectively)	Randomized controlled clinical trial	Postpartum Depression and Anxiety	EPDS, SSTAI	27 mg Zn-sulfate or 320 mg Mg-sulfate/die	8 weeks	No significant difference in EPDS and SSTAI scores between groups.	Mg and zinc did not reduce postpartum anxiety and depressive symptoms	(−)
[81]	71 medication-free ADHD (55 M, 16 F; mean age 9.7 ± 1.5)	randomized clinical trial	ADHD	ADHD-RS-IV, CGI, CGAS	Broad spectrum micronutrient formula (Daily Essential Nutrients, DEN) up to 12 cp/die, vs placebo (no Mg doses available)	10 weeks	Most children entered the trial with nutrient levels falling within expected ranges. Regression analyses showed varying predictors across outcomes with no one of the predictors being consistently identified across different variables.	Limited value of using serum nutrient levels to predict treatment response, although Authors cannot rule out that other non-assayed nutrient levels may be more valuable	(±)
[82]	264 (69 M, 195 F; mean age 31.6±8.5)	Randomized, single-blind clinical trial	Depressive, anxiety and stress symptoms	DASS-42	Mg–vitamin B6 combination, respectively 300 mg/die and 30 mg/die or Mg alone (Magnespasmyl) 300 mg/die	8 weeks	Both treatment arms reduced DASS-42 stress subscale score from baseline to Week 8. Adults with high stress score had a 24% greater improvement with Mg-vitamin B6 versus Mg at Week 8.	Mg supplementation alleviated stress in healthy adults with hypomagnesemia. Addition of vitamin B6 to Mg was not superior to Mg supplementation alone	(+)
[83]	12 patients (3 M, 9 F, mean age 46.5±9)	Double-blind crossover trial	Treatment-Resistant Depression	PHQ-9, HAMD	4 g Mg-sulfate/die in 5% dextrose or placeboinfusion of 5% dextrose	8 days intervention periods with a 5 days washoutin between periods	No changes were recorded on the HAMD or PHQ-9 24 h post- treatment; serum Mg increased from baseline to day 7, PHQ-9 decreased from baseline to day 7.	Intravenous infusion of Mg- sulfate did not reduce depressive symptoms in adults with treatment-resistant depression	(−)

Captions: (PHQ-9) Patient Health Questionnaire-9; (GAD-7) Generalized Anxiety Disorders-7; (EPDS) Edinburgh Postnatal Depression Scale; (SSTAI) Spielberger State-Trait Anxiety Inventory; (ADHD-RS-IV) ADHD Rating Scale IV; (DASS-42) Depression Anxiety Stress Scales; (CGAS) Children Global Assessment Scale; (+) positive outcome; (−) negative outcome; (±) unclear outcome.

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
