# Peer review of "The Role and the Effect of Magnesium in Mental Disorders: A Systematic Review"

_nutrients, 2020, doi:10.3390/nu12061661_

Round 1

Reviewer 1 Report

The study presents a nicely executed review of the possible effects of Mg and mental health disorders. I do have some comments, but all in all, I read the study with interest

  1. Abstract - a few minor typos
  2. Introduction - is well written but the positioning of the 1.1.-1.4 is not clear. Why are they presented? how the sub-themes were selected (nb 1.4 - there is a typo it became 2.4). So my suggestion is to elaborate a bit more on why these points are separately discussed, especially as they seem to be at times repetitive to what was said in the introduction
  3. Methods - the study does not fully justify the use of term systematic. I would suggest using the term comprehensive review. The origanisation of the section presenting the search process can be more systematic. i.e. the search terms can be better organised and presented by themes, I miss the whole search syntax, please add it. Please explicify the inclusion and exclusion criteria and make it clear who of the authors did the search (add initials), was there a cross-validation of the selection? What was the agreement rate? what was the possibility of the bias? MAJOR POINT in methods: how were the data extracted and analysed? the whole section on that is totally missing.
  4. Results - MAJOR POINT I miss the whole section on what kind of papers were selected, so to say 32 papers at a glace, what were they about? when were they published? how big the sample sizes were etc. Why do you need to present each disorder you came across? Readers know what depression is and how prevalent it is. I would pay a lot more attention to the presentation of the findings of the relavent papers: you identified 12 studies for depression. currently, the presentation of the findings is quite lengthy and is mostly a presentation of each study in detail. while some details are needed, I would move majority of the info in the table per disorder (expand the one you have already) and make a summative analysis of the papers instead of going paper by paper. Suggestion: split results tables in 2 or 3 sub-tables. In one present the methodological details and maybe main conclusions, in the others go more in-depth of the findings.
  5. Discussion and conclusion - are quite short and I feel are not discussing the strengths and limitations of this study sufficiently. Also the link to other previous studies in the field is quite superficial.

Author Response

COMMENTS TO THE AUTHORS

Milan, May 13th 2020

To the kind attention of

Prof. Paolo Brambilla and Prof. Agostoni Carlo, Guest Editors, Special Issue entitled " Nutrients and Brain across the lifespan", Nutrients

Thank you for considering our manuscript entitled “The role and the effect of Magnesium in mental disorders: a systematic review" by Botturi et al. for publication at Nutrients.

We are grateful to you for the most helpful comments which have all been taken into account and that have helped us to greatly improve our manuscript, which has now been accordingly re-structured.

In particular, as per reviewer #1 input, we re-structured our manuscript by also reducing the introduction and adding the information requested to the method section. Moreover, we also have modified the discussion and the organization of each table of manuscript as suggested by the reviewer#1. Finally, all authors carefully re-read the manuscript for English style and grammatical errors.   

All the changes made to the original manuscript in response to all the reviewer suggestions have been addressed point by point.

We look forward to receiving your final decision on our manuscript.

Sincerely,

Dr. Botturi Andrea on behalf of all the authors

Neurologic Clinic, Fondazione IRCCS Istituto neurologico Carlo Besta, 20133 Milan, Italy, e-mail: andrea.botturi@istituto-besta.it

Reviewer#1: 
General:
The study presents a nicely executed review of the possible effects of Mg and mental health disorders. I do have some comments, but all in all, I read the study with interest

RESPONSE: We thank the reviewer for summarizing our study. All the reviewer's comments have been taken into consideration and, we believe, have greatly ameliorated the quality of the revised version of the manuscript. We thank the referee for sharing her/his thoughts and having spent her/his precious time on reviewing the manuscript.

Details

Abstract - a few minor typos

RESPONSE: We thank the referee for this input. We have carefully reread the abstract and we have corrected minor typos, that now you can read as follow:

“Introduction: Magnesium is an essential cation involved in many functions within the central nervous system, including transmission and intracellular signal transduction. Several studies have shown its usefulness in neurological and psychiatric diseases. Furthermore, it seems that magnesium levels are lowered in the course of several mental disorders, especially depression. Objectives: To evaluate the presence of a relationship between the levels of magnesium and the presence of psychiatric pathology as well as the effectiveness of magnesium as a therapeutic supplementation. Methods: Systematic search of scientific records concerning magnesium in psychiatric disorders published from 2010 up to March 2020 was performed. We collected a total of 32 articles (18 on Depressive Disorders, 4 on Anxiety Disorders, 4 on Attention Deficit Hyperactivity Disorder, 3 on Autism Spectrum Disorder, 1 on Obsessive-Compulsive Disorder, 1 on Schizophrenia and 1 on Eating Disorder). Results: Twelve studies highlighted mainly positive results in depressive symptoms. Seven showed a significant correlation between reduced plasma magnesium values and depression measured with psychometric scales. Two papers reported improving depressive symptoms after taking magnesium intake, two in association with antidepressants, compared to controls. No significant association between magnesium serum levels and panic or GAD patients in two distinct papers was found. In other two paper reduced score in HAM-A in depressed patients correlate with more high levels of magnesium and beneficial of magnesium in stressed patients was showed. Two paper reported low levels of magnesium in association with Attention Deficit Hyperactivity Disorder (ADHD). Only one of three paper showed lower levels of magnesium in Autism Spectrum Disorder. Eating Disorder and schizophrenia reported variation of magnesium levels in some aspects of the disease. Conclusion: The results are not univocal both in terms of the plasma levels and of therapeutic effects. However, from the available evidence, it emerged that supplementation with magnesium could be beneficial. Therefore, it is necessary to design ad hoc clinical trials to evaluate the efficacy of magnesium alone or together with other drugs (antidepressants), to establish the correct use of this cation with potential therapeutic effect.”

Introduction - is well written but the positioning of the 1.1.-1.4 is not clear. Why are they presented? how the sub-themes were selected (nb 1.4 - there is a typo it became 2.4). So my suggestion is to elaborate a bit more on why these points are separately discussed, especially as they seem to be at times repetitive to what was said in the introduction

RESPONSE: Thank you very much for pointing this out. Since magnesium exerts different function in each human organ, we decided to insert sub-themes aiming at describing not only the specific role and effect of magnesium on brain functions in psychiatric disorders but also its effect on psychopharmacological medications. However, we fully agree with the reviewer’s comment and some sub-paragraphs have been deleted and the remaining one have been reduced. Therefore, the introduction section now reads as follows:

Psychiatric disorders are estimated to affect more than 1 billion people worldwide [1] and recent reports claimed that global burden of mental illness accounts for as high as one-third of years lived with disability (YLDs) and 13% of disability-adjusted life years (DALYs)[2]. These figures are associated with a high economic burden since the global direct and indirect economic costs of mental disorders are estimated at $2.5 trillion [3].

The therapeutic options for treating psychiatric disorders are progressively expanding, with both pharmacologic and non-pharmacologic treatments becoming available in the last years. Moreover, although the majority of drugs currently used to treat the more common psychiatric disorders have been proven to be effective by recent robust meta-analyses [4,5], most of the psychiatric pharmacotherapy must be continued for years (or even lifelong), bringing severe adverse side effects [6,7]. Furthermore, the last decade has seen a steep increase in the price of both brand and generic psychiatric prescription drugs [8] and therefore there is great need of new therapeutic options that prove to be effective, safe and affordable for the patient and the healthcare system.

Oral nutritional supplements have been shown to improve clinical outcomes of hospitalized patients [9] and to be cost-effective interventions [10]. In recent years, there has been a surge in the number of studies performed in the field of the so-called Nutritional Psychiatry [11] where researchers have been focusing not only on the effects of general dietary modifications on some psychiatric illnesses, in particular mood disorders [12], but also on the potential role of supplementation of single micronutrients in patients with mental disorders [13]. The most substantial pieces of evidence on the efficacy of these interventions are probably those coming from omega-3 polyunsaturated fatty acids (PUFAs) supplementation in depression [14,15], which, in few years, will likely become part of the standard therapy for depression [16].

Among micronutrients, magnesium (Mg2+) plays a critical role in brain function and mood since it is essential for optimal nerve transmission and it is involved in the formation of membrane phospholipids. For this reason, it plays a fundamental role in the correct functioning of the central nervous system.

Specifically, for psychiatric illnesses, multiple studies have been performed in which magnesium serum levels were assessed in cohorts of patients with depression [17], schizophrenia [18], addiction disorders [19], anxiety disorders [20] and ADHD [21].

Magnesium has also been studied, both in the form of enriched diet and supplementation at high doses, as an adjunct therapy for psychiatric disorders, in particular in anxiety [22] and mood [23] disorders. However, results from reports on both magnesium levels and supplementation in psychiatric illnesses are often conflicting, potentially due to methodological heterogeneity, which involves, among others, measuring techniques (extracellular vs ionized magnesium) [24] and supplementation modalities (dose, posology, magnesium form used).

Since magnesium could represent a potentially novel adjunctive therapy in mental disorders, with this review, we aim to give a comprehensive picture of the relationship between magnesium and psychiatric disorders. In particular, we will try to answer the following questions: a) Are magnesium levels altered in patients with psychiatric disorders? b) Is magnesium supplementation effective and safe in patients suffering from mental illnesses? If so, is it possible to identify a preferred dose, posology or element form?

To answer these questions, we performed a comprehensive review in which we included all studies that involved patients with a wide variety of psychiatric disorders, in which magnesium levels were assessed and/or magnesium supplementation was given.

1.1. Biological Plausibility of Magnesium for brain and psychiatric disorders

Magnesium is the fourth most abundant mineral ion, and the intake comes mainly from the ingestion of leafy green vegetables, whole grains, nuts, and fish. Magnesium is absorbed in the gastrointestinal tract and the renal system. This element facilities calcium (Ca2+) absorption and both ions are regulated by parathyroid hormone; however, free ion concentration does not always correlate with total concentration. Magnesium is primarily found within the cell, while extracellular magnesium accounts for 1% of total body magnesium. Also, serum magnesium is present in three forms, with ionized magnesium having the highest biological activity [25].

Magnesium is essential to ensure the correct functioning of all human cells, neurons included; it is involved, among others processes, in hundreds of enzymatic reactions [26], intracellular transmission [27], myelination process [28], synapses formation and maintenance [29] as well as in the regulation of serotoninergic, dopaminergic and cholinergic transmission [30]. Magnesium is, therefore, an element necessary to maintain neurons healthy and viable [31], especially because it has been shown to reduce apoptosis in an animal model of induced hypoxia-ischemia [32] and to prevent synaptic loss in a mouse model of Alzheimer disease [33].  Shreds of evidence also suggest that magnesium is involved in neurogenetic processes and the maturation of newly generated neural cells; in fact, magnesium has been proven to efficiently enhance the proliferation of neural stem cells [34] and neurite outgrowth [35]. Through induction of synaptic plasticity and potentiation of synaptic transmission in the rat hippocampus, magnesium has also been shown to enhance learning abilities, working memory and short- and long-term memory [36].

The antidepressant action of magnesium is likely to be mediated by several mechanisms. The most important one seems to be the one involving the blockade of the glutamatergic N- methyl-D-aspartate receptor (NMDAR); interestingly, this is the same target of the fast-acting antidepressant ketamine, an NMDAR antagonist too. However, other components of the glutamatergic transmission, such as the AMPA receptor, appear to be also modulated by magnesium [37]. Another relevant finding that supports the notion of magnesium antidepressant activity is the one linking magnesium deficiency to dysregulation in the hypothalamic-pituitary- adrenal (HPA) axis, that is well known to be involved in the pathogenesis of anxiety disorders and depression [38]. Conversely, increased levels of brain magnesium have been shown to enhance a) the retention of the extinction of fear memory, through increased NMDA signalling, b) the brain-derived neurotrophic factor (BDNF) expression and c) synaptic plasticity in the prefrontal cortex (PFC); of note, these effects were absent in another region closely linked to depression pathogenesis, such as the basolateral amygdala [39].

The antidepressant action of magnesium appears to be, at least partially, mediated by a modulation of the serotoninergic system; in fact, it seems that magnesium has a synergistic effect when administered with molecules of the selective serotonin reuptake inhibitors (SSRI) class and that the antidepressant action of magnesium is impaired when animals were pre-treated with a compound that inhibits the serotonin synthesis [40]. Interestingly, in rats, a diet deficient in magnesium were also found to be associated with alterations in the gut microbiota, ultimately leading to alterations in the gut-brain axis and the development of depressive-like behaviours [41]. Finally, Sowa-Kućma et al. [42] found that concentrations of both magnesium and zinc (Zn) were reduced in the hippocampal tissue of suicide victims, along with an altered glutamatergic NMDA activity in the hippocampus. Moreover, several reports suggest that magnesium is a key mediator of the efficacy of antipsychotic medications. Both haloperidol and risperidone were found to increase intra-erythrocytic magnesium levels [43] and treatment of hypoparathyroidism-induced psychosis seems to be dependent on magnesium level, with hypomagnesemia causing treatment resistance to antipsychotics [44]. Magnesium also appears to be involved in both prevention and reversal of movement disorders induced by the chronic use of typical antipsychotics. Moreover, in an animal model, magnesium was found to reduce the severity of movement disorders via the prevention of the formation of reactive oxygen species in cortical areas, striatum and substantia nigra [45].

Methods - the study does not fully justify the use of term systematic. I would suggest using the term comprehensive review. The origanisation of the section presenting the search process can be more systematic. i.e. the search terms can be better organised and presented by themes, I miss the whole search syntax, please add it. Please explicify the inclusion and exclusion criteria and make it clear who of the authors did the search (add initials), was there a cross-validation of the selection? What was the agreement rate? what was the possibility of the bias? MAJOR POINT in methods: how were the data extracted and analysed? the whole section on that is totally missing.

RESPONSE: We thank the reviewer for allowing us to clarify this point. In accordance to reviewer’s input, we included in the manuscript the whole search syntax in the supplementary materials in order to provide a more systematic organization of our search process, by depicting the different phases of our search by specifying the initials of the name of authors that carried out the search, and by describing how they have reached the agreement rate. Finally, we have included in the manuscript a specific section describing how data were extracted and analyzed. The whole methods section has been totally restructured and reads as follow:

A comprehensive search of all studies using or analysing the effect of magnesium in psychiatric disorders were conducted on the PubMed database from January 1, 2019 to March 30, 2020. Search was re-run on a weekly basis, with the last search performed on April 10, 2020. Finally, we searched for potentially valuable records by scanning reference lists of articles relevant to the topic. The final search strategy for PubMed was designed by AB and the search syntax is reported in the supplementary materials.

Articles of potential interest were identified by using the following search strategy: “(magnesium OR Mg2++) AND (psychiatric disorders OR mental diseases OR psychotic disorders OR psychosis OR ultra-high risk psychosis OR schizophrenia OR bipolar disorder OR affective disorder OR major depressive disorder OR depression OR  personality disorder OR anxiety disorders OR obsessive compulsive disorders OR eating disorders OR ADHD OR autism)”. Only studies in English (or with an English translation available) were taken into consideration. Relevant articles were obtained and included in the review if (a) they reported an exposure to magnesium; (b) included psychiatric symptoms as an outcome measure; and (c) enrolled human participants and reported a trial or an observational study. We considered cohort studies exploring serum levels of magnesium as primary outcome and trials in which the authors used an exposure of magnesium alone or as an adjunctive therapy to other drugs (e.g., antipsychotic, antidepressants, mood stabilizers and benzodiazepines), or other no pharmacological strategies, such as psychotherapy and physical exercise compared to placebo or pharmacotherapy alone. In addition, we included trials that employed a diet enriched in magnesium as a supplementation. All the studies, both clinical trial and cohort studies (prospective or retrospective) reporting effect of magnesium in patients were included.

To limit the heterogeneity of this review and to reduce selection biases, we decided to exclude: 1) studies that did not explore the effects of magnesium on psychiatric symptoms as primary outcome; 2) pre-clinical studies, both in vitro and in vivo (animal); 3) case-control, case series or case reports; 4) reviews and/or metanalysis.

Search results were exported into the reference manager software “Rayyan QCRI”; duplicates detected by the software were resolved manually by (AB).  After duplication removal, all the resulting records were screened by title and abstract by AB and BV and initially labeled for inclusion with “included”, “excluded” or “maybe”. Inclusion of records labeled with “maybe” was discussed between the two reviewers, with other members of the team involved if needed. On the second level of screening, full text of publications was evaluated by AB and BV; disagreements on study selection were resolved by consensus with the involvement of a third author (VC).Two reviewers (AB and BV) independently charted the data and discussed the results; any disagreement was resolved by consensus with the involvement of a third team member (VC). Since the search was re-run on a weekly basis, data from newly included study were updated accordingly. Then data items collected were analyzed according to the following variables that were abstracted for each article (see supplementary materials).

The results were grouped based on the observational or clinical trial investigating the effect of magnesium on patients’ mental health. Findings were discussed on the basis of different diagnosis made (e.g. depression, schizophrenia) and were summarized in a table form. Results for each clinical topic were presented in a narrative form in the main text.

We identified 1104 citations by searching the PubMed database. After duplicates were removed a total of 698 records were considered for inclusion. After the first screening based on the title and abstract, 621 records were excluded, with 77 full text articles to be retrieved and assessed for eligibility. After this second screening process, 45 studies were excluded for the following reasons: 11 were animal studies; 7 not reported effect of magnesium as primary outcomes; 18 not reported as results the serum levels of magnesium; 3 are reviews or metanalysis; 4 are preclinical studies and 2 not investigated adults sample. Finally, a total of 33 articles were considered eligible for this review. The process of identification and inclusion of trials is summarized in Figure 1. (PRISMA diagram).”

Supplementary materials

SEARCH SYNTAX:

magnesium AND (psychiatric disorders OR mental diseases OR psychotic disorders OR psychosis OR ultra-high risk psychosis OR schizophrenia OR bipolar disorder OR major depressive disorder OR afffective disorder OR depression OR personality disorder OR anxiety disorder OR obsessive-compulsive disorder OR eating disorders OR ADHD OR autism)

clinical studies:

  • Classical Article
  • Clinical Study
  • Clinical Trial
  • Clinical Trial Protocol
  • Clinical Trial, Phase I
  • Clinical Trial, Phase II
  • Clinical Trial, Phase III
  • Clinical Trial, Phase IV
  • Comparative Study
  • Controlled Clinical Trial
  • Dataset
  • Editorial
  • Evaluation Study
  • Guideline
  • Introductory Journal Article
  • Journal Article
  • Letter
  • Multicenter Study
  • Observational Study
  • Overall
  • Practice Guideline
  • Pragmatic Clinical Trial
  • Randomized Controlled Trial
  • Review
  • Validation Study

Temporal criterion for strategy: past 10 years.

VARIABLES abstracted for each article included:

  • General information (with sub-columns)
  • Title
  • First author
  • Country of origin
  • Date of publication
  • Publication status (preprint record, published on peer-reviewed journal)
  • Study design
  • Clinical topics covered by the study (with sub-columns)
  • psychiatric disorders”/ “mental diseases”,
  • “psychotic disorders”/ “psychosis”/ “ultra-high risk psychosis”/ “schizophrenia”,
  • “bipolar disorder”
  • “major depressive disorder”/ “affective disorder”/ “depression”,
  • “personality disorder”
  • “anxiety disorders”
  • “obsessive compulsive disorders”
  • “eating disorders”,
  • “ADHD”
  • “autism”

Results - MAJOR POINT I miss the whole section on what kind of papers were selected, so to say 32 papers at a glace, what were they about? when were they published? how big the sample sizes were etc. Why do you need to present each disorder you came across? Readers know what depression is and how prevalent it is. I would pay a lot more attention to the presentation of the findings of the relavent papers: you identified 12 studies for depression. currently, the presentation of the findings is quite lengthy and is mostly a presentation of each study in detail. while some details are needed, I would move majority of the info in the table per disorder (expand the one you have already) and make a summative analysis of the papers instead of going paper by paper. Suggestion: split results tables in 2 or 3 sub-tables. In one present the methodological details and maybe main conclusions, in the others go more in-depth of the findings.

RESPONSE: Thank you for allowing us to clarify this point. In accordance to reviewer’s input, we have extensively modified the Results section, as the reviewer suggested, which addresses this insightful comment and reads as follows:

Most of the results of the reviewed studies focused on depression and depressive symptoms and only a small number of studies concerning other psychiatric disorders were found.

4.1. DEPRESSION

Risk factors for depression include dietary patterns. Some epidemiological or observational studies reported that greater dietary intake of Mg is linked to a general reduced risk of depressive disorders or fewer depressive symptoms [49,50]. However, some studies also suggested that in depressive disorders Mg plasma levels can show different or synergistic effects [51]. To clarify this issue, in this paragraph we reviewed all RCT studies exploring the impact of Mg on depressive disorder. First, we explored the RCT studies (Table 1) exploring Mg levels in depressed patients. We identified twelve studies showing mainly positive results, even though they were conducted in different populations and measuring symptoms with various tools.

A group of studies investigated several microelements, including serum Mg, in depressed patients vs healthy controls. Most studies reported a significant decreased in concentrations of Mg in depressed patients [52],[53],[54],[55],[56] whereas two studies [57],[58], conducted in a female population, reported no significant decrease of Mg levels.

We identified three studies [59], [60], [61], that correlated Mg levels with severity of symptoms, measured with a variety of tools, reporting, though, mixed results.

One study [59] showed only a small correlation between psychomotor retardation and Mg plasma levels. Interestingly, when patients were divided between responder and non-responder to treatment (SSRI or SNRI), they demonstrated that patients with higher plasma Mg levels at baseline improved better compared to that with lower Mg levels at baseline.

Finally, we identified two studies conducted on healthy populations, evaluating the correlation between Mg levels and depressive symptoms. Specifically, Tarleton et al., (2019) [62] considered Patient Health Questionnaire (PHQ) scores in a large cohort of 3604 healthy adults and showed a significant relationship between serum Mg and symptoms. Similarly, Salehi-Pourmehr et al., (2019) [63], which conducted a study on overweight pregnant women, also reported a positive significant correlation.

Furthermore, we identified five RCT studies (Table 2 and Table 3) in which Mg was administered alone or as add-on treatment to depressed patients, reporting, also in this case, contrasting results. Specifically, we found three positive studies showing the efficacy of Mg supplementation in the treatment of depression [64]; [65];[66]. In contrast, Fard et al. (2017) [67] showed that Mg did not reduce anxiety and depressive symptoms in postpartum women. In the same direction, Mehdi et al. (2017) [68] did not find a significant effect of Mg sulfate in affecting depression.

Finally, a single, smaller study [69] addressing the efficacy and safety of Mg as an augmentation to antidepressant treatment, found no significant differences.

In conclusion, current evidence on the impact of Mg on depression should be supported using longitudinal studies with extended follow up, larger sample sizes and repeated psychopathological evaluations at different times.

4.2. OTHER PSYCHIATRIC DISORDERS

Anxiety Disorders

In our records screening, we identified four studies that analysed the concentration of Mg in patients suffering from anxiety disorders. Three of them reported no significant differences in Mg serum levels in Generalized Anxiety Disorder (GAD), Panic Disorder and anxiety symptoms during a major depressive episode ([70];[71];[59]). In contrast, in Camardese et al. (2012) [59], the authors hypothesized that hypomagnesaemia could play a role in drug-responsiveness among depressed patients, as they found a correlation between lower Mg levels and poor outcome in treated patients.

Garalejić et al. (2010) [72] also investigated the relationship between Hamilton Anxiety Scale (HAMA) scores and Mg levels, although, unlike other studies identified, this study considered Mg levels in peritoneal fluid among 87 infertile women undergoing laparoscopy, finding a strong negative correlation between the severity of anxiety symptoms and Mg concentration in peritoneal fluid. Therefore, the authors hypothesized that the decrease in Mg peritoneal concentration may be caused by a higher production of endogenous catecholamines (predominantly adrenaline) in patients showing higher scores at the anxiety symptoms assessment.

Finally, as for the use of Mg supplementation in psychopharmacological treatment, we found one study that considered effects of Mg supplementation on anxiety symptoms [67]. As depicted above, Fard et al. (2017) [67] explored both the baseline trace elements level and possible benefits of Zn and Mg supplements on depression and anxiety symptoms among postpartum women. Anxiety symptoms were evaluated using the Spielberger State-Trait Anxiety Inventory and no statistically significant difference was observed in mean scores of state anxiety and trait anxiety.

Obsessive-Compulsive Disorder (OCD)

There are very few scientific data about serum levels of microelements in OCD patients. Indeed, we found only one study carried out by Shohag et al. (2012) [73] that described a decrease in Mg levels, together with Zn and Fe levels, in OCD patients when compared to healthy controls.

Schizophrenia

We found a total of one study concerning Mg and SCZ (Table 1). Specifically, Jabotinsky‐Rubin et al. (1993) [74] reported that patients with SCZ had increased Mg plasma concentrations and also that Mg levels were reduced after administration of haloperidol. In contrast, Athanassesnas et al., (1983) [75] reported no differences in plasma Mg concentrations in drug-free patients with SCZ. Moreover, in our records screening, we found one study, carried out by Ruljancic et al. (2013) [76], that studied Mg concentration in suicidal and non-suicidal patients with SCZ, which reported a higher Mg concentration in the platelets of suicidal patients and a higher Ca2+/Mg2+ ratio in the platelets of non-suicidal patients, an indirect index of higher Ca2+ concentration. However, the imbalance found in the two electrolytes in the platelets of suicidal and non-suicidal patients with SCZ still requires further research, in order to clarify the involvement of Mg and Ca2+ in SCZ, as well as a possible relation to antipsychotic action.

Eating Disorders

A large retrospective study by Raj et al. (2012) [77] aimed to determine the prevalence of hypomagnesemia (Mg2+ ≤1.7 mg/dL) and clinical characteristics of adolescents hospitalized with a DSM-IV-diagnosis of eating disorder. As expected, they found hypomagnesemia in approximately 16% of eligible participants. Also, the authors found that compared with those with normal serum Mg levels, patients with hypomagnesemia were older, with longer duration of illness, more likely to be purging, and more likely to have an alkaline urine. However, they did not differ in eating disorder diagnosis, Body Mass Index, or other electrolyte disturbances.

Attention-Deficit/Hyperactivity Disorder (ADHD)

In recent years, some studies [78] started to explore the association between micronutrient dietary intake and ADHD development. Specifically, Rucklidge et al. (2019) [79] (Table 2 and Table 3) conducted a parallel–group RCT to assess the efficacy and safety of a broad-spectrum micronutrient formula compared with placebo in 93 medication-free children diagnosed with ADHD. With regard to micronutrients levels among the subjects, most children entered the trial with nutrient blood levels falling within expected ranges. Data analysis only resulted in not statistically significant associations between micronutrient treatment response and pre-treatment serum nutrient levels, showing the limited value of using serum nutrient levels to predict treatment response.

Moreover, Yang et al. (2019) [80] (Table 1) conducted a study with the aim to investigate the trace element status of Zn, Cu, Fe, Mg and lead in children with ADHD and healthy controls. They enrolled Chinese children diagnosed with ADHD according to DSM-5 criteria [81], under the following three presentations: predominantly inattentive presentation, predominantly hyperactive/impulsive presentation and combined presentation; they were between the ages of 6 and 16 years and had no history of psychopharmacological treatment for their condition. The authors found that there were alterations in blood levels of Zn in ADHD patients, which was associated with their symptom scores. However, unlike Zn levels, the study failed to find a correlation between Mg and ADHD diagnosis or symptom scores.

In contrast, in a case-control study, Mahmoud et al. (2011) [82] found that Mg levels were significantly lower in children with ADHD compared to controls. Finally, in a very recent publication, Skalny et al. (2020) [83] also observed a significant difference in Mg between ADHD patients and gender- and age-matched neurotypical controls. However, the patterns of trace element and mineral levels in ADHD were significantly affected by gender and age.

Autism Spectrum Disorder (ASD)

Regarding trace elements role in this group of disorders, two studies did not find a statistically significant difference in levels of Mg in children diagnosed with ASD compared to age-matched and gender matched neurotypical children ([84];[85]) (Table. 1), while one study [86], demonstrated lower levels of Mg in a large cohort of Chinese children diagnosed with ASD.

Finally, in order to better clarify the point raised by the reviewer we also have split the Table in 3 different tables. Table 1,2 showed sample size, type of study, methodological details whereas Table 3 showed more details on findings related to RCT studies (e.g. results, outcomes, directions of evidences). All tables are modified and inserted in the manuscript.

Discussion and conclusion - are quite short and I feel are not discussing the strengths and limitations of this study sufficiently. Also the link to other previous studies in the field is quite superficial.

RESPONSE: We thank the reviewer for this suggestion. In accordance to reviewer’s input, we included different paragraphs in the Discussion section, which addresses this insightful comment and reads as follows:

“The aim of this review was to provide a comprehensive overview of the effects of magnesium in different psychiatric disorders. Interestingly, from the reviewed studies emerged that the results showing the association between mental disorders and magnesium are still largely inconclusive. Specifically, we found a great number of studies evaluating serum levels of magnesium in different mental disorders, especially depression. However, only few RCTs were testing the efficacy of magnesium alone or as added therapy in the treatment of different psychiatric disorders and only two studies explored the presence of magnesium in dietary habits of schizophrenic and depressed population, respectively. Notably, the presence of many studies on depression is not surprising since the use of magnesium is mostly reserved to depressive disorders, because of its involvement in a several number of core mechanisms of depressive physiopathology, including glutamatergic transmission in the limbic system and cerebral cortex [79], regulation of the HPA axis [80], inflammation and oxidative stress [81], response to NMDA receptor antagonist [82], serotonin, dopamine, and noradrenaline modulation [83], BDNF expression[37], as well as modulation of sleep–wake cycle [84]. More in details, previous evidence reported that the potential efficacy of magnesium in depression may be linked to the modulation of glutamatergic signals, which play a key role on neuroprotection, and to the fact that magnesium acts as antagonist on NMDA receptors [40]. Moreover, evidence also showed that magnesium may have a synergistic effect combined to antidepressant. Indeed, the review carried out by Serefko et al. (2016) suggested that magnesium could improve the efficacy of standard antidepressant treatments, and as such, could be an add on treatment to the standard antidepressant [84]. The role of magnesium in depression has been also demonstrated in several preclinical studies. Interestingly, Poleszak et al. (2005) found that magnesium enhanced the antidepressant effect of imipramine in mice using forced swim test (FST)[85]. Additionally, the same research group, in 2006, showed that combining sub-therapeutic doses of Mg2+ in combination with sub-therapeutic doses of imipramine leads to a significant antidepressive like effect in animal models [86]. Also, Singewald et al. (2004) demonstrated that imipramine could be able to reverse depression like behaviour in rats with low levels of magnesium [87]. In addition, Poleszak et al. (2007), demonstrated that magnesium administered in combination to an NMDA antagonist, called MK-801, which is similar to ketamine, amplify its antidepressant effect [88]. More recently, Murck et al.(2013) reported that magnesium and ketamine showed an overlap of action in animal models because of both of them could lead to synaptic sprouting[89], ultimately suggesting that they both have a similar action in SNC. By the way the authors suggest that in depressed patients magnesium levels could be used to predict the effect of ketamine[89].

Therefore, based on this evidence showing the key role of magnesium in influencing mechanisms that may conduct to depression, further study investigating the impact of antidepressant drugs on intracellular magnesium concentration in neurons are required.   

Interestingly, the research carried out on depression highlighted the linked between the development of this disabling illness and reduced plasmatic levels of magnesium, evidence that is line with previous reviews and meta-analyses [23]. In the light of these results, several studies [52, 53, 54, 58] suggested that, for adults seen in primary care, lower serum magnesium levels were associated with depressive symptoms, ultimately supporting the use of supplemental magnesium as therapy. For this reason, magnesium could be considered a hallmark of pathology or could represent a biomarker of response to drug treatment in patients with mood disorders, as also reported by a previous review [52]. Indeed, patients with therapy refractory depression appear to have lower CNS Magnesium levels in comparison to health controls [54]. In a prospective of developing a nutrition therapies for depressed patients with lower levels of magnesium could be useful make a dosage of this mineral and combined to the standard antidepressant treatment magnesium to ameliorate the outcomes of disease, following a personalized approach to depression.

However, only a handful of studies investigated the efficacy of magnesium supplementation alone or as add on therapy  to other drugs Specifically, from these studies emerged that magnesium alone [61,57,58], magnesium and others micronutrients [66, 72], magnesium with vitamin B6 [59] or magnesium in combination with antidepressants [62] could not be considered significantly effective for treating depression, since the results are conflicting. Interestingly, Medhi et al. (2017)[61], was the only study that administered un intravenous infusion of magnesium and reported that this formulation had only a partial, but not significant, anti-depressive effect in depressed patients. A possible explanation could be due to the particular pharmacokinetics of magnesium. Indeed, the total concentration of magnesium is mainly intracellular and free ion concentration does not always correlate with whole concentration.

Importantly, comorbidities and other confounders, such as age and geographic location, may contribute to the discrepant findings.  In addition, other factors, linked to psychiatric patients, could impact on these findings, such as sedentary life style, unhealthy dietary patterns, smoke, alcohol and substances abuse, lower compliance to treatments.

Similarly, also the evidence reporting the relationship between anxiety disorders and magnesium are still conflicting, although this association is well established in the scientific literature [85]. However, these negative results might be due to the small number of studies investigating magnesium values in anxiety disorders in the past 10 years, ultimately suggesting the need of future research focusing on elucidating magnesium's mechanism of action in order to determine if it has anxiolytic properties.

Furthermore, mixed results have been also reported by studies investigating the link between dietary pattern and deficiency of magnesium or other minerals (e.g. Zn, Fe) in ADHD patients, with some studies showing an efficacy and other not. However, it is possible that these heterogenous results could be linked to different minerals supplemented, which did not permit to examine the real effect of each minerals and also to different characteristics of patients enrolled in the original studies, which were not homogeneous in terms of age, severity and subtype of ADHD.

Finally, due to the paucity of studies investigating ASD, SCZ and OCD, at present is not possible to determine the role of magnesium in either the physiopathology or in the treatment of these disorders. Similarly, regarding eating disorders, we found only one study where the presence of various internal conditions seems to mask the pathophysiological role of magnesium. This represent an unexplored field and could be interesting elucidating the role and effect of magnesium also in these disorders.

Importantly, this review might be considered in the light of some limitations derived from the heterogeneity of the included studies in terms of (1) types of magnesium supplementation (2) target population (3) follow-up period, (4) outcome measures, (4) severity of illness (5) sample sizes, (6) comorbidities, (7) life style. All these factors may have limited the generalizability of the results and made it difficult to compare the results emerged from the available studies.

In conclusion, due to the lack of consistency between the available studies, there is limited evidence that magnesium, alone or as add-on therapy, is useful in treating different psychiatric disorders, even if several data showed reduced plasma levels, especially in depressive patients. Therefore, larger and homogenous studies are required for showing the role and the effects of magnesium in psychiatric illnesses.”

Reviewer 2 Report

I read with much interest the study presented in manuscript nutrients-786291 aiming to systematically review the evidence of a relationship between circulating levels and intake  of magnesium with the presence of psychiatric disorders and to review the evidence of effectiveness of magnesium as a therapeutic supplementation. The paper concerns a topic of much interest in the field of nutrition confirmed by the extensive literature available on this topic. The authors conclude that results are not uniform in the two aims of the review, but some evidence suggest a possible benefit, which justifies the implementation of well-design clinical trials in order to establish the correct use of magnesium in psychiatric conditions.The review is comprehensive, updated, the references are adequate and the article is generally clear. The conclusions are in accordance with the attentive review.  I believe that this review is very informative and worthy of publication. The only point I would raise is to proofread the manuscript for few typo and English grammar mistakes.

Author Response

COMMENTS TO THE AUTHORS

Milan, May 13th 2020

To the kind attention of

Prof. Paolo Brambilla and Prof. Agostoni Carlo, Guest Editors, Special Issue entitled " Nutrients and Brain across the lifespan", Nutrients

Thank you for considering our manuscript entitled “The role and the effect of Magnesium in mental disorders: a systematic review" by Botturi et al. for publication at Nutrients.

We are grateful to you for the most helpful comments which have all been taken into account and that have helped us to greatly improve our manuscript, which has now been accordingly re-structured.

In particular, as per reviewer #1 input, we re-structured our manuscript by also reducing the introduction and adding the information requested to the method section. Moreover, we also have modified the discussion and the organization of each table of manuscript as suggested by the reviewer#1. Finally, all authors carefully re-read the manuscript for English style and grammatical errors.   

All the changes made to the original manuscript in response to all the reviewer suggestions have been addressed point by point.

We look forward to receiving your final decision on our manuscript.

Sincerely,

Dr. Botturi Andrea on behalf of all the authors

Neurologic Clinic, Fondazione IRCCS Istituto neurologico Carlo Besta, 20133 Milan, Italy, e-mail: andrea.botturi@istituto-besta.it

Reviewer#2: 
I read with much interest the study presented in manuscript nutrients-786291 aiming to systematically review the evidence of a relationship between circulating levels and intake of magnesium with the presence of psychiatric disorders and to review the evidence of effectiveness of magnesium as a therapeutic supplementation. The paper concerns a topic of much interest in the field of nutrition confirmed by the extensive literature available on this topic. The authors conclude that results are not uniform in the two aims of the review, but some evidence suggest a possible benefit, which justifies the implementation of well-design clinical trials in order to establish the correct use of magnesium in psychiatric conditions. The review is comprehensive, updated, the references are adequate and the article is generally clear. The conclusions are in accordance with the attentive review.  I believe that this review is very informative and worthy of publication. The only point I would raise is to proofread the manuscript for few typo and English grammar mistakes.

RESPONSE: We thank the referee for appreciating our review. We thank the referee for sharing her/his thoughts and having spent her/his precious time on reviewing the manuscript.  In accordance to reviewer's input, the English language and the typos have been carefully revised and polished.